# The Modulation of Functional Status of Bovine Spermatozoa by Progesterone

**DOI:** 10.3390/ani11061788

**Published:** 2021-06-15

**Authors:** Vitaly Denisenko, Irena Chistyakova, Natalia Volkova, Ludmila Volkova, Baylar Iolchiev, Tatyana Kuzmina

**Affiliations:** 1Branch of Federal Research Center for Animal Husbandry Named after Academy Member L.K. Ernst, Russian Research Institute of Genetic and Breeding Farm Animals, 196601 Saint-Petersburg, Russia; den.vitaly2016@yandex.ru (V.D.); itjerena7@gmail.com (I.C.); 2Federal Research Center for Animal Husbandry Named after Academy Member L.K. Ernst, 142132 Moscow, Russia; natavolkova@inbox.ru (N.V.); ludavolkova@inbox.ru (L.V.); baylar1@yandex.ru (B.I.)

**Keywords:** progesterone, bull spermatozoa, capacitation, acrosome reaction

## Abstract

**Simple Summary:**

Progesterone is an endogenous steroid hormone, which can induce capacitation and/or acrosome reactions in semen of certain mammalian species. Our study aimed to investigate the effect of progesterone on the functional status of fresh bovine spermatozoa using a chlortetracycline fluorescent probe. Results showed that heparin induced capacitation in spermatozoa incubated with or without progesterone. The destruction of microfilaments by an inhibitor of cytochalasin D blocked the stimulating effect of heparin. Steroid hormone in mixture with prolactin stimulated the acrosome reaction in spermatozoa, which was blocked by an inhibitor of microtubule polymerization (nocodazole). At the acrosome stage, prolactin provided the undergoing of acrosome reaction in male gametes. This effect was noted both in the presence and absence of progesterone and inhibited by nocodazole. The supplementation of dibutyryl cyclic adenosine monophosphate during the acrosome reaction to progesterone-untreated spermatozoa did not cause changes in proportion of acrosome-reacted cells. However, when progesterone was added during capacitation, a significant increase in the proportion of capacitated cells was noted, which was inhibited by nocodazole. Thus, progesterone under the action of prolactin and dibutyryl cyclic adenosine monophosphate determines the functional status of fresh spermatozoa, which indicates progesterone-modulating effect on the indicators of post-ejaculatory maturation of male gametes.

**Abstract:**

The aim of this study is to identify the effects of progesterone (PRG) on the capacitation and the acrosome reaction in bovine spermatozoa. The fresh sperm samples were incubated with and without capacitation inductors (heparin, dibutyryl cyclic adenosine monophosphate (dbcAMP)), hormones (prolactin (PRL), PRG), inhibitors of microfilaments (cytochalasin D) and microtubules (nocodazole) during capacitation and acrosome reactions. The functional status of spermatozoa was examined using the chlortetracycline assay. Supplementation of heparin stimulated capacitation in the presence and absence of PRG. Cytochalasin D blocked the stimulating effect of heparin on capacitation. The addition of PRL during capacitation (without PRG) did not affect the functional status of spermatozoa, while in PRG-treated cells PRL stimulated the acrosome reaction. PRL (with and without PRG) increased the acrosome reaction in capacitated cells. These PRL-dependent effects were inhibited by nocodazole. During the acrosome reaction, in presence of dbcAMP, PRG decreased the proportion of acrosome-reacted cells compared to PRG-untreated cells. This effect in PRG-treated cells was canceled in the presence of nocodazole. In conclusion, PRG under the action of PRL and dbcAMP determines the changes in the functional status of native sperm cells, which indicates PRG modulating effect on the indicators of post-ejaculatory maturation of spermatozoa.

## 1. Introduction

Capacitation is a biochemical process, which includes changes in membrane proteins and lipids, ion fluxes, an increase in the level of cyclic adenosine monophosphate (cAMP) and protein phosphorylation [1]. Successful fertilization of the oocyte includes many biomechanical and biochemical changes in spermatozoa when passing the oviductial tract and reaching the oocyte-cumulus complex [2]. Progesterone (PRG) is the main physiological activator of acrosome reaction, which is a Ca^2+^-dependent process [3]. Progesterone is secreted by cumulus cells and present in high concentration during ovulation in the follicular fluid, where it can act on sperm cells before they bind to the zona pellucida [4]. The follicular fluid promotes the processes of capacitation and acrosome reaction in bull spermatozoa [5]. Progesterone is an endogenous steroid hormone, which can induce capacitation and/or acrosome reaction in semen of certain mammalian species, and sperm response to the progesterone is highly species-specific. It has been well established that progesterone induces the capacitation or acrosome reaction in horse and human spermatozoa [6,7], but the data on the effect of the progesterone on bovine sperm cells are contradictory. In some cases, the progesterone provides only the capacitation, or only the acrosome reaction in pre-capacitated spermatozoa, or induces both cellular processes [8,9,10]. It has been shown that the acrosome reaction occurs only in capacitated cells [8]. Progesterone activates the capacitation process; however, it has been noted that PRG-induced acrosome reaction also can happen [11]. Progesterone-induced changes are mediated by intracellular mechanisms associated with protein kinase C and voltage-gated Ca^2+^ channels [12]. By stimulating or inhibiting protein kinases A, C, G and tyrosine kinase, it has been shown that progesterone induces the acrosome reaction via tyrosine kinase and protein kinase C, but in horse sperm cells this process does not depend on protein kinases A, C and bicarbonate [13]. In contrast, in human sperm, the acrosome reaction is dependent on protein kinase A [14]. Calcium is also an important modulator of the capacitation and the acrosome reaction, probably participating as a key mediator in the exchange of information between the sperm and the egg [15]. Induction of the acrosome reaction by progesterone leads to an increase in calcium entry from the extracellular environment. The control of the function of voltage-gated Ca^2+^ channels, apparently, helps to prevent the premature acrosome reaction [16]. In bovine spermatozoa, heparin and dbcAMP are compounds that activate processes of capacitation [17,18], while PRL induce the acrosome reaction [19]. One of the main processes of capacitation is the dynamic transformation of the cytoskeleton, in particular, actin. Actin is the most widely known cytoskeleton protein, acting as a secondary messenger in signal transduction [20]. Actin polymerization occurs during capacitation in various mammalian species, including cattle [20,21]. Microtubules are also involved in intracellular processes that determine the acrosome reaction [19].

The aim of this study is to identify the effects of different hormones (progesterone, prolactin) on the postejaculate processes (capacitation and acrosome reaction) in bovine spermatozoa, and the role of microtubules and microfilaments in these modifications.

## 2. Materials and Methods

### 2.1. Chemicals

All chemicals were obtained from Sigma-Aldrich Co. (Steinheim, Germany), unless otherwise indicated.

### 2.2. Location

The present study was performed in the development biology laboratory of the All-Russian Research Institute of Genetics and Farm Animal Breeding of the Federal Research Center for Animal Husbandry, named after academy member L.K. Ernst (Saint Petersburg-Pushkin, 59°42′35″ N, 30°27′5″ E).

### 2.3. Semen Collection and Preparation

One ejaculate was collected from each of the sixty Black-and-White bulls (aged 3–4 years old) just before the experiment. All sperm samples were derived from animals of breeding farm OJSC Nevskoe immediately after ejaculation. Fresh sperm was collected twice weekly, with no apparent changes in animal health or semen quality throughout the semen collection interval. Fresh ejaculates from three bulls were mixed and the obtained mixture was used for each experiment.

In order to remove the seminal plasma, the obtained fresh sperm was twofold centrifuged at 300× *g* for 10 min in Sp-TALP medium consisting of 100 mM NaCl, 3.1 mM KCl, 25 mM NaHCO_3_, 0.3 mM NaH_2_PO_4_, 21.6 mM sodium lactate, 0.5 mM CaCl_2_, 0.4 mM MgCl_2_, 10 mM HEPES, 1 mM pyruvate and 0.1% polyvinyl alcohol (PVA, 30,000–70,000 Da).

### 2.4. Study Design

#### 2.4.1. Induction of Capacitation

After sperm dilution to a final concentration 50 × 10^6^ sperm/mL, fresh samples were divided into sixteen equal aliquots. Eight aliquots were incubated in Sp-TALP medium with supplementation of 6 mg/mL bovine serum albumin (BSA), 0.5 mM CaCl_2_ and with and without different reagents-capacitation inductor (heparin), hormones (PRL, PRG) and inhibitors of actin microfilaments (cytochalasin D) and tubulin microtubules (nocodazole): 1—without any reagents (control), 2—with 5 µg/mL heparin (or 10 ng/mL PRL) [18,22]; 3—with 5 µg/mL heparin (or 10 ng/mL PRL) and 10 µM cytochalasin D [23]; 4—with 5 µg/mL heparin (or 10 ng/mL PRL) and 10 µM nocodazole [24]; 5—with 1 mg/mL PRG (control) [7]; 6—with 1 mg/mL PRG, 5 µg/mL heparin (or 10 ng/mL PRL); 7—with 1 mg/mL PRG, 5 µg/mL heparin (or 10 ng/mL PRL) and 10 µM cytochalasin D; 8—with 1 mg/mL PRG, 5 µg/mL heparin (or 10 ng/mL PRL) and 10 µM nocodazole. The incubation was performed in the atmosphere of 5% CO_2_ at 38.5 °C and 95% humidity for 4 h.

#### 2.4.2. Induction of Acrosome Reaction

Acrosome reaction in spermatozoa, preliminary capacitated with 5 µg/mL heparin, was performed using 100 µg/mL lysophosphatidylcholine [25] and incubated in Sp-TALP medium with supplementation of 6 mg/mL BSA, 0.5 mM CaCl_2_ and with and without different reagents-capacitation inductor (dibutyryl cyclic adenosine monophosphate, dbcAMP), hormones (PRL, PRG) and inhibitors of actin microfilaments (cytochalasin D) and tubulin microtubules (nocodazole): 1—without any reagents (control), 2—with 100 µM dbcAMP (or 10 ng/mL PRL) [17]; 3—with 100 µM dbcAMP (or 10 ng/mL PRL) and 10 µM cytochalasin D; 4–with 100 µM dbcAMP (or 10 ng/mL PRL) and 10 µM nocodazole; 5—with 1 mg/mL PRG (control); 6—with 1 mg/mL PRG, 100 µM dbcAMP (or 10 ng/mL PRL); 7—with 1 mg/mL PRG, 100 µM dbcAMP (or 10 ng/mL PRL) and 10 µM cytochalasin D; 8—with 1 mg/mL PRG, 100 µM dbcAMP (or 10 ng/mL PRL) and 10 µM nocodazole. The incubation was performed in the atmosphere of 5% CO_2_ at 38.5 °C and 95% humidity for 30 min (Figure 1).

### 2.5. CTC Fluorescence Assay

Chlortetracycline (CTC) was dissolved at 750 µM in a buffer of 20 mM Tris, 130 mM NaCl and 5 mM L-cysteine, and the pH was adjusted to 7.8. The solution was kept in a light-shielded container at 4 °C. At the time of the assay, 20 μL of the sperm suspension of each sample was mixed with 20 μL of the CTC solution and incubated with CTC at 38.5 °C for 10 min. Then, 10 μL of 25% glutaraldehyde in 1 mM Tris (pH 7.4) was added to a final concentration of 0.1% glutaraldehyde in each sample for fixation. A drop of sperm suspension (10 μL) was placed on a glass slide and mixed with 10 μL of 0.22 M 1,4-diazobicyclo [2.2.2] octane dissolved in glycerol/phosphate buffered saline (9:1) at room temperature. Then, coverslip was attached, drops of colorless nail polish were applied to fix its edges. The slides were stored in a light-shielded container at 4 °C before scoring.

The slides were scored with a ZEISS AxioLab. A1. (Karl Zeiss, Jena, Germany) equipped with phase contrast and epifluorescence optics. The excitation for CTC was 380–400 nm, the emission–530 nm. The samples were analyzed in accordance with one of the three types of CTC fluorescence patterns [26]: the uniform fluorescence in the entire sperm head with uncapacitated cells, the fluorescence-free band in the post-acrosome region; capacitated cells, the absence of fluorescence in the entire sperm head, with the exception of a thin bright band of fluorescence in the equatorial area; and acrosome-reacted cells (Figure 2).

### 2.6. Statistical Analysis

Results of the present study are predominantly presented using descriptive statistics. Average percent of sperm cells (in capacitation stage and acrosome reaction stage) in control and experimental groups were compared by Student’s *t*-test, with data presented as means ± SEM. Results were considered significant when *p* < 0.05; *p* < 0.01; *p* < 0.001. Statistical analysis of the results was carried out using Statistic 7.0 package (StatSoft, Tulsa, OK, USA, 2016).

## 3. Results

### 3.1. The Effect of PRG on Functional Status of Bull Spermatozoa during Capacitation

The analysis of effects of 1 µg/mL PRG on the capacitation process of bull spermatozoa stimulated by heparin (5 µg/mL) is presented in Figure 3. The use of the chlortetracycline probe allows for visualization of the redistribution of Ca^2+^ in the sperm membrane by forming CTC-Ca^2+^-membrane-bound fluorescence complexes. The sperm cells, which were incubated for 4 h with heparin showed an increase of capacitated cells (26%, *p* < 0.001) and a decrease of acrosome-reacted spermatozoa (69%, *p* < 0.001) compared to intact untreated cells incubated for 4 h (16% and 81%, respectively, *p* < 0.001). The treatment with an inhibitor of microfilaments (cytochalasin D) at the concentration of 10 µM resulted in a lower percent of capacitated cells and a high proportion of acrosome-reacted sperm cells (12% and 83%, respectively, *p* < 0.001) versus those of cells treated with heparin only. The treatment of sperm with 10 µM inhibitor of microtubules (nocodazole) did not affect the cell ratio with different fluorescence pattern (25% vs. 69%, respectively, *p* < 0.001) compared to the group incubated with heparin. It should be noted that the effects of a single use of heparin or a mixture with PRG on spermatozoa were similar—an increase in the number of capacitated cells and a decrease in acrosome-reacted spermatozoa (26% and 23% vs. 69% and 70%, respectively). The usage of the inhibitor cytochalasin D stimulated the decrease of acrosome-reacted cells in PRG-treated group of spermatozoa during capacitation (83%, *p* < 0.01).

The effect of PRL (10 ng/mL) on appearance of bovine spermatozoa with different fluorescence patterns is shown in Figure 4. No significant changes in the ratio of sperm cells with different functional status was observed during the incubation period of cells with 10 ng/mL PRL (without PRG) for 4 h in comparison with capacitation of spermatozoa without PRL. The treatment of cells with cytochalasin D in PRG-free medium promoted the growth in number of acrosome-reacted cells during the 4 h incubation of spermatozoa with PRL in comparison with the sperm cells incubated 4 h with any reagents (82% vs. 63%, respectively, *p* < 0.001). The treatment with PRG caused changes in the biological action of PRL on sperm cells during incubation period; as well, the stimulatory effect on the growth of acrosome-reacted sperm cells was blocked by the microtubule inhibitor, nocodazole (81% and 82% vs. 70%, respectively, *p* < 0.01).

### 3.2. The Effect of PRG on Functional Status of Bull Spermatozoa during Acrosome Reaction

The effect of 10 ng/mL PRL on the time course of acrosome reaction in bovine spermatozoa is shown in Figure 5. The incubation of cells with PRL, without PRG, during the acrosome reaction period resulted in a high percent of acrosome-reacted cells (81%, *p* < 0.001) in comparison with intact and PRG-treated control groups (70% and 67%, *p* < 0.001). Furthermore, the PRL-induced effect on stimulation of the acrosome reaction process in spermatozoa was inhibited by nocodazole (70%, *p* < 0.001) versus PRL-treated PRG-untreated group (81%, *p* < 0.001). The incubation of sperm cells with PRL or the mixture with PRG, stimulated an undergoing of acrosome reaction in capacitated spermatozoa (81% and 82%, respectively) in contrast with both control groups, which was canceled under the action of nocodazole (70% and 69%, respectively, *p* < 0.001).

The effect of PRG on the acrosome reaction process activated by 100 μM dbcAMP presented in Figure 6. The addition of dbcAMP to intact PRG-untreated sperm cells had no effect on the time course of the acrosome reaction (78%, respectively, *p* < 0.001) compared to intact and PRG-treated control groups (78% and 78%, respectively). The pre-incubation of cells with PRG and subsequent exposure to dbcAMP increased the number of capacitated cells compared to the spermatozoa incubated with dbcAMP only (29% vs. 19, respectively, *p* < 0.001). During incubation of PRG-treated spermatozoa with the cytoskeleton-depolymerized agent cytochalasin D, no significant changes in the ratio of sperm cells with a different functional status caused by exposure to dbcAMP were observed (25% and 68% and 29% and 64%, respectively, *p* < 0.001), whereas the nocodazole promoted the increase of acrosome-reacted sperm cells (15% and 78%, respectively, *p* < 0.001).

## 4. Discussion

In our experiments, in the absence of PRG, heparin and dbcAMP stimulated the capacitation and did not affect the process of acrosome reaction in fresh sperm. PRL worked in the opposite way—it increased the proportion of acrosome-reacted spermatozoa and did not have an effect on cells at the capacitation stage.

During capacitation, actin polymerization occurs in mammalian spermatozoa. Before the acrosome reaction, actin-cleaving proteins are activated, which leads to actin depolymerization [27]. Actin polymerization is one of the mechanisms dependent on capacitation that protects sperm from premature acrosome reaction [28]. Polymerization of microtubules occurs during the acrosome reaction, which is necessary for the normal fertilization process [29]. It can be assumed that the action of compounds, which activate the passage of capacitation, should be associated with the polymerization of actin; compounds that stimulate the polymerization of microtubules are suitable to stimulate the acrosome reaction. According to our results, the heparin-stimulated capacitation is associated with the functioning of actin cytoskeleton, since the depolymerization of microfilaments mediated by cytochalasin D, caused the decrease in the proportion of capacitated sperm cells. In buffalo spermatozoa during capacitation, dbcAMP also stimulates actin polymerization [30]. Microtubules are necessary for implementation of effect of PRL during the acrosome reaction period, since the simultaneous use of the inhibitor nocodazole led to the dramatic drop in the proportion of acrosome-reacted cells in bovine spermatozoa. In breast cancer cells, prolactin promotes actin regulation, thus enhancing cell movement [31]. Also, microtubules can participate in the transmission of the prolactin signal from the receptor to the genes of milk protein [32]. The usage of homological follicular fluid for incubation of bovine spermatozoa in various concentrations, containing a large amount of PRG, promoted sperm capacitation and induced the acrosome reaction [5]. However, the data on the effect of PRG on the acrosome reaction are controversial. It has been shown that under certain conditions, PRG can stimulate the acrosome-associated processes, although there are significant differences between the data presented in the articles of different researchers [33,34].

Thus, in intact PRG-untreated bovine spermatozoa, capacitation activators (heparin, dbcAMP) increase the number of capacitated cells only during process of capacitation, and microfilaments are involved in this process. At the same time, an increase in the number of acrosome-reacted cells under the action of PRL occurred only when spermatozoa underwent the acrosome reaction with intact microtubules.

In intact (PRG-untreated) spermatozoa of bulls, PRL did not promote processes of acrosome reaction during capacitation, while in sperm cells exposed with PRG, PRL activated the acrosome reaction in capacitated spermatozoa. In intact bovine spermatozoa, the effect of dbcAMP at the stage of the acrosome reaction had no effect on proportions of spermatozoa with different functional status, whereas in the presence of PRG, dbcAMP increased the percent of cells on capacitation stage. The action of dbcAMP during capacitation in untreated sperm cells was associated with microfilaments; the effect of PRL during the acrosome reaction was mediated by microtubules. At the same time, the effect of PRL at the stage of capacitation and the effect of dbcAMP during the acrosome reaction, in the presence of PRG, depended only on the intact microtubules.

Thus, in bovine spermatozoa treated with PRG, the effects of dbcAMP and PRL did not depend on functional status of male gametes (capacitation or acrosome reaction). In this case (treatment with PRG) dbcAMP always (both at the capacitation stage and at the stage of acrosome reaction) increased the percent of capacitated spermatozoa, while PRL always (at both stages) stimulated the growth in the proportion of acrosome-reacted spermatozoa.

After thawing bull sperm cells, their fertility is significantly reduced. The lower fertility of cryopreserved semen is partly due to premature capacitation-like modifications in considerable proportion of the sperm that affects the sample quality [35]. It has also been shown that cryopreservation procedures destroys microfilaments in mammalian cells and does not affect microtubules [36]. This finding raises the question of what the possible mechanism is for showing additional capacitated sperm cells after thawing. This increase likely occurs through the “intracellular mechanisms”, potentially due to the transformation of spermatozoa from capacitation stage to acrosome-reacted. In the presence of PRG in fresh bovine spermatozoa, all compounds, which activate processes of capacitation (heparin and dbcAMP) and acrosome reaction (prolactin), act through the microtubules, which remain intact even after thawing (Figure 7).

## 5. Conclusions

It can be assumed that, in thawed cell samples, an increase in the number of capacitated spermatozoa occurs due to the transition of acrosome-reacted cells to capacitated ones. According to our results obtained in fresh sperm cells, the induction of acrosome reaction in capacitated sperm cells in thawed samples may be mediated by the action of PRG or another compound acting in a similar manner. Furthermore, this cell transformation is associated with the functioning of cytoskeleton.

## Figures and Tables

**Figure 1 animals-11-01788-f001:**
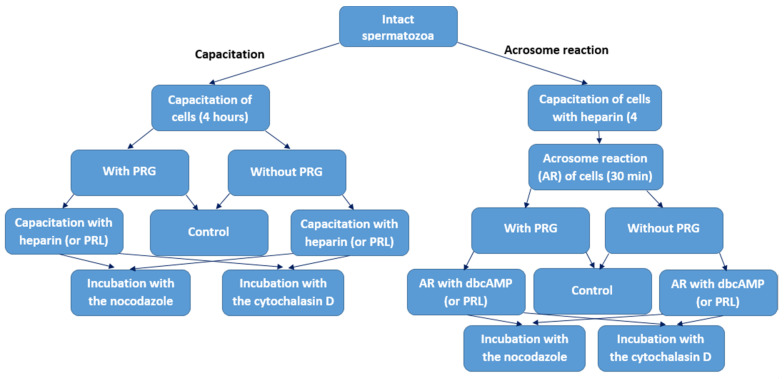
Study design.

**Figure 2 animals-11-01788-f002:**
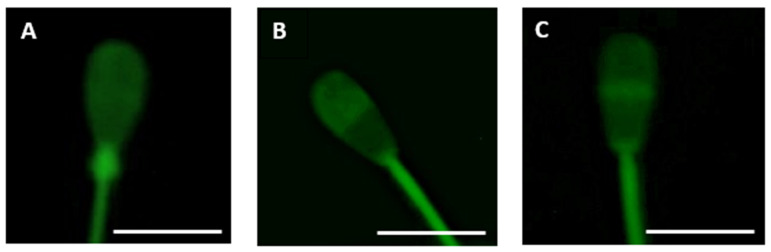
Fluorescence intensity of CTC-Ca^2+^-membrane-bound complex in bull spermatozoa was detected by CTC fluorescence probe. (**A**) non-capacitated sperm cell; (**B**) capacitated sperm cell; (**C**) acrosome-reacted sperm cell. Scale bar: 7 µm. The fluorescence patterns of a minimum of 200 sperm were scored in each experimental group. All slides were kept at room temperature in light-shielded containers until examination. The sample on the slide was divided into quadrants with 6–8 separate fields being examined per quadrant.

**Figure 3 animals-11-01788-f003:**
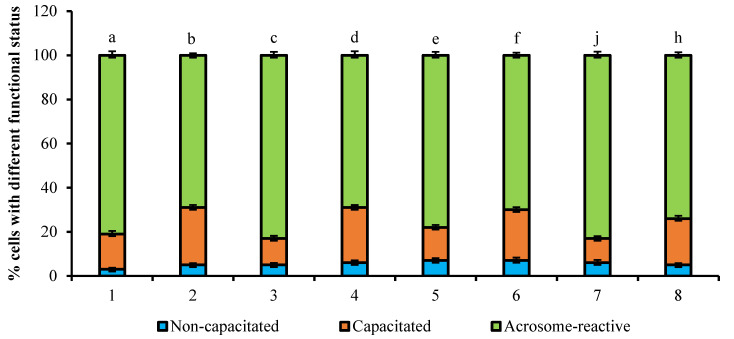
The effect of 1 µg/mL PRG on capacitation process of bull spermatozoa stimulated by heparin (5 µg/mL) (total number of counted cells-6400; number of experiments-4). 1—without any reagents (control), 2—with 5 µg/mL heparin; 3—with 5 µg/mL heparin and 10 µM cytochalasin D; 4—with 5 µg/mL heparin and 10 µM nocodazole; 5— with 1 mg/mL PRG (control); 6—with 1 mg/mL PRG, 5 µg/mL heparin; 7—with 1 mg/mL PRG, 5 µg/mL heparin and 10 µM cytochalasin D; 8—with 1 mg/mL PRG, 5 µg/mL heparin and 10 µM nocodazole. The differences are significant (percent ratio of capacitated and acrosome-reacted cells) ^a:b; a:d; b:c; c:d; c:b; b:j; d:j; a:f; c:f; f:j^ at *p* < 0.001; ^b:d; d:e; a:h; e:f; f:j; e:h; j:h^ at *p* < 0.01 (Student’s *t*-test).

**Figure 4 animals-11-01788-f004:**
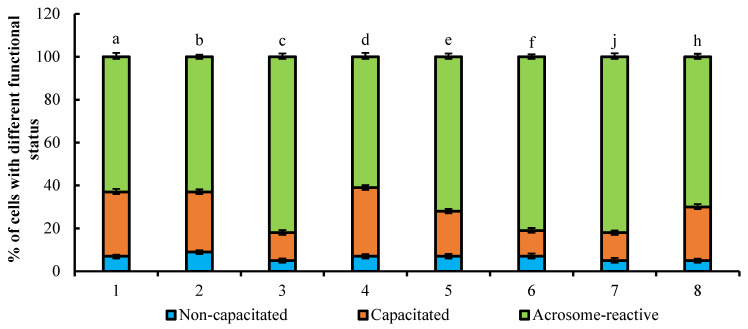
The effect of PRL (10 ng/mL) on appearance of bovine spermatozoa with different fluorescence patterns (total number of counted cells-6400; number of experiments-4). 1—without any reagents (control); 2—with 10 ng/mL PRL; 3—with 10 ng/mL PRL and 10 µM cytochalasin D; 4—with 10 ng/mL PRL and 10 µM nocodazole; 5—with 1 mg/mL PRG (control); 6—with 1 mg/mL PRG, 10 ng/mL PRL; 7—with 1 mg/mL PRG, 10 ng/mL PRL and 10 µM cytochalasin D; 8—with 1 mg/mL PRG, 10 ng/mL PRL and 10 µM nocodazole. The differences are significant (percent ratio of capacitated and acrosome-reacted cells) ^a:c; a:b; c:d; a:f; a:j; b:c; b:f; b:j; d:c; d:f; d:j; h:f; h:j^ at *p* < 0.001; ^d:e; a:e; a:h; b:e; b:h; d:h^ at *p* < 0.01 (Student’s *t*-test).

**Figure 5 animals-11-01788-f005:**
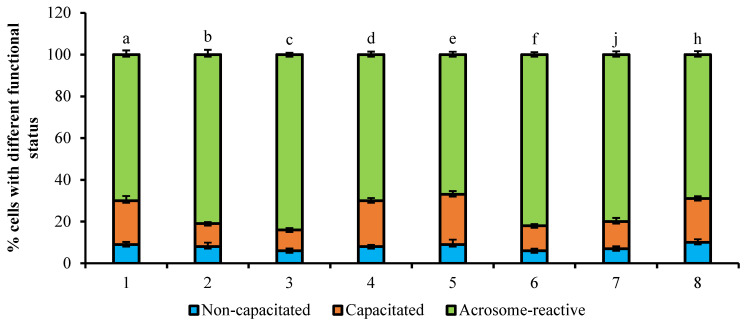
The effect of 10 ng/mL PRL on the time course of acrosome reaction in bovine spermatozoa (total number of counted cells-6400; number of experiments-4). 1—without any reagents (control); 2—with 10 ng/mL PRL; 3—with 10 ng/mL PRL and 10 µM cytochalasin D; 4—with 10 ng/mL PRL and 10 µM nocodazole; 5—with 1 mg/mL PRG (control); 6—with mg/mL PRG, 10 ng/mL PRL; 7—with 1 mg/mL PRG, 10 ng/mL PRL and 10 µM cytochalasin D; 8—with 1 mg/mL PRG, 10 ng/mL PRL and 10 µM nocodazole. The differences are significant (percent ratio of capacitated and acrosome-reacted cells) ^a:b; a:c; a:f; a:j; d:b; d:e; d:f; d:j; e:b; e:c; e:f; e:j; h:b; h:e; h:f; h:j^ at *p* < 0.001 (Student’s *t*-test).

**Figure 6 animals-11-01788-f006:**
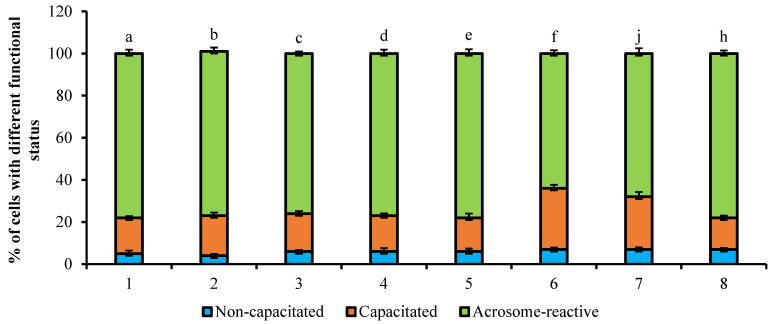
The effect of PRG on the acrosome reaction process activated by 100 μM dbcAMP (total number of counted cells-6400; number of experiments-4). 1—without any reagents (control); 2—with 100 µM dbcAMP; 3—with 100 µM dbcAMP and 10 µM cytochalasin D; 4—with 100 µM dbcAMP and 10 µM nocodazole; 5—with 1 mg/mL PRG (control); 6—with mg/mL PRG, 100 µM dbcAMP; 7—with 1 mg/mL PRG, 100 µM dbcAMP and 10 µM cytochalasin D; 8—with 1 mg/mL PRG, 100 µM dbcAMP and 10 µM nocodazole. The differences are significant (percent ratio of capacitated and acrosome-reacted cells) ^a:f; b:f; c:f; d:f; e:f; h:f^ at *p* < 0.001; ^a:j; b:j; c:j; d:j; e:j; h:j^ at *p* < 0.01 (Student’s *t*-test).

**Figure 7 animals-11-01788-f007:**
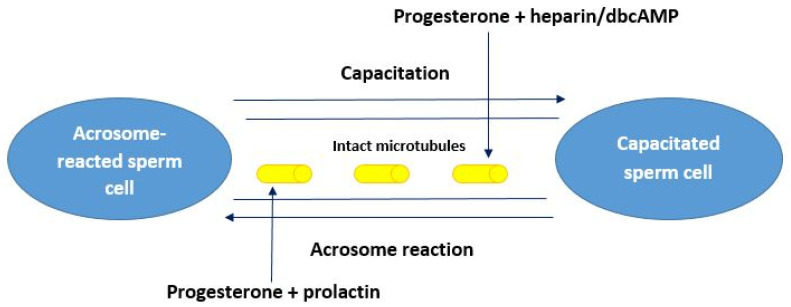
The effect of progesterone on functional status of native bovine spermatozoa. In the presence of progesterone in native bovine spermatozoa, dbcAMP and heparin induce the process of capacitation in spermatozoa at the acrosome reaction stage, while prolactin stimulate the acrosome reaction in capacitated cells; both reactions go through the intact microtubules.

## Data Availability

Not applicable.

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
