# Peer review of "The Modulation of Functional Status of Bovine Spermatozoa by Progesterone"

_animals, 2021, doi:10.3390/ani11061788_

Round 1
Reviewer 1 Report
page 2, line 80-81: clarify please the status of bovine spermatozoa (fresh or thawed)?
page 2, line 95: Ejaculates from three bulls werw pooled or not?
At the section of study design (page 3) it could be better to introduce subtitles at the beginning of each paragraph. At the first paragraph the title of pragraph could be Induction of capacitation and at the second one Induction of acrosome reaction.
Also, in the figure 1 could be add these subtitles at the top of each diagram.
page 8, line 282: the authors mention the use of bull sperm cells after thawing and the effect of progesterone on functional status, as well as at figure 5. In my opinion, it is necessary to define the method of freezing-thawning of spermatoza which used and the kind of cryoprotectants.
Also, in the conclusions you assumed for thawed cell samples which in the design of the study is not mentioned! It could be better to carry out an experiment with the use of cry-preserved semen samples.
Author Response
Good afternoon!
We appreciate the time and effort that you has dedicated to providing your valuable feedback on our manuscript. We are grateful to you for their insightful comments on our paper. We have highlighted the changes within the manuscript.
Also let us give some explanations:
Point 1: Thank you for pointing this out. We have added information about status of bovine spermatozoa in the revised version of our manuscript to emphasize this point. We used in our experiment only fresh bovine sperm.
Point 2: Yes, three ejaculates were pooled. We have, accordingly, added information in our manuscript.
Point 3: We agree with this comment. Therefore, we have changed the section «study design» (and also the figure) according to your recommendations.
Point 4: In experiments, we used fresh semen samples. The mention of the thawed semen was made for possible extrapolation of our results obtained from native spermatozoa on cryopreserved ones. In the addition, we got some interesting results from this point.
Point 5: As we said earlier, we used for experiments only fresh semen samples.
We look forward to hearing from you in due time regarding our submission and to respond to any further questions and comments you may have.
Reviewer 2 Report
This manuscript characterize the effects of progesterone on capacitation and acrosome reaction in bull sperm. In view of the controversy findings on the role of progesterone in sperm maturations, it is very interesting to get knew knowledge.
Please find attached my suggestions.

Author Response
Good afternoon!
We appreciate the time and effort that you has dedicated to providing your valuable feedback on our manuscript. We are grateful to you for their insightful comments on our paper. We have been able to incorporate changes to reflect most of the suggestions provided by you. We have highlighted the changes within the manuscript.
Also let us give some explanations about further sections:
Material and methods:
Point 1: Age of the animal, the season and the duration of the experiment - Thank you for pointing this out. The average age of bulls was (3-4 years old), the season – summer-autumn, the duration of the experiment – 3 months
Point 2: The viability and the motility before performing the assay – Thank you for pointing this out. The motility of spermatozoa was estimated (42-43 % of sperm cells with progressive motility), the viability also was estimated after thawing (45-50 % of viable cells).
Point 3: Presentation of the results - Thank you for this suggestion. However, in the case of our study, we tried to make tables, but it looked very cumbersome and incomprehensible.
Discussion:
In experiments, we used fresh semen samples. The mention of the thawed semen was made for possible extrapolation of our results obtained from native spermatozoa on cryopreserved ones. In the addition, we got some interesting results from this point.
We look forward to hearing from you in due time regarding our submission and to respond to any further questions and comments you may have.
Reviewer 3 Report
This Reviewer regrets to say that this manuscript has serious lacks in originality and does not present novel results.
The references are not updated at all. Please, revise the works published during the last years regarding bovine spermatozoa and progesterone effects. This is a mandatory condition for this Reviewer prior to revise again this manuscript.
English languange should be revised, also in terms of style.
Specific comments:
-Figure 1: low quality, not readable.
-Regarding the counting of the spermatozoa, I would suggest to use the automatic counting by programs such as Image J, which randomnly choose the fields, avoind the bias of the individual performing the count.
-It is strongly suggested to revise the statistical analysis.
-How can the Authors affirm and stablish these conclusions with only fluorescence analysis on 200 spermatozoa? It is not clear for this Reviewer.
Author Response
Good afternoon!
We appreciate the time and effort that you has dedicated to providing your valuable feedback on our manuscript. We are grateful to you for their insightful comments on our paper. We have highlighted the changes within the manuscript.
Also let us give some explanations:
Point 1: The novelty of our research, in our opinion, is based on complex methods of using fluorescent and inhibitory analysis to determine the effects of progesterone and prolactin on post-ejaculatory maturation of the male gametes with a clear fixation of functional status of spermatozoa (visualization of the localization of membrane-bound calcium), what allow us to obtain the data on the involvement of elements of the cytoskeleton in the action of hormones for spermatozoa, depending on their functional state.
Point 2: Thank you for pointing this out. We have updated the list of references of our manuscript according to your recommendation.
Point 3: Thank you for pointing this out. We have tried to revise the style of language and make some corrections of all sections.
Also the answers for specific comments:
Point 4: Thank you for pointing this out. We have tried to make another scheme, more readable.
Point 5: Thank you for this suggestion. However, we suppose that eye method more reliable, more over one sample was counted by two-three different people, that negate the bias.
Point 6: Can you specify the failures of the statistical analysis, which you mean please? Thank you.
Point 7: Thank you for pointing this out. We have scored 200 cells of each experimental group (there were 8) in 4 repetitions (the total number was 6400 cells). Also we have added information about scoring of sperm cells.
We look forward to hearing from you in due time regarding our submission and to respond to any further questions and comments you may have.
Round 2
Reviewer 1 Report
The authors should be stated clear that the used semen at their experiments was fresh.
Author Response
Good afternoon!
We are grateful to you for further comment on revised version of our manuscript. We have been able to make some changes according to your recommendation. We have highlighted the changes within the manuscript.
Thank you so much!